# Dynamics of Low-Lying Sandy Coast of the Gydan Peninsula, Kara Sea, Russia, Based on Multi-Temporal Remote Sensing Data

**Nataliya Belova [1], Alexander Ermolov [1], Anna Novikova [1,\*], Stanislav Ogorodov [1] and Yulia Stanilovskaya [2]**

[1] Laboratory of Geoecology of the North, Faculty of Geography, Lomonosov Moscow State University, 1 Leninskie Gory, 119991 Moscow, Russia
[2] TotalEnergies, 2 Jean Miller, La Defense, 92078 Paris, France
\* Correspondence: anna.novikova@geogr.msu.ru; Tel.: +7-495-939-2526

**Abstract:** The retreat rates of Arctic coasts have increased in recent decades at many sites, and an essential part of coasts considered accumulative before have turned erosional due to global climate changes and construction in the coastal zone. In this paper, we study a 7 km long coastal section of the western Gydan Peninsula in a new construction area. Based on the interpretation of multi-temporal satellite imagery, we assessed coastal dynamics in distinct periods from 1972 to 2020. We analyzed the geological structure of the coast as well as changes in hydrometeorological parameters with time, and considering the human impact, we proposed the main drivers of spatial and temporal variations of coastal dynamics. The studied low-lying sandy accumulative marine terrace was more or less stable in the period before construction (1972–2014). However, with the area's development, the coast dynamics changed drastically: in 2014–2017, three-quarters of the studied area experienced retreat, and the average retreat rate amounted to 5.8 m/yr, up to 28.5 m/yr near the construction sites. We relate this coastal erosion intensification to human impact combined with the growth of hydrometeorological forcing. Although coastal erosion slowed down after 2017, the retreat trend remained. In the coming years, with Arctic climate warming, erosion of the studied coast will continue.

**Keywords:** Arctic coasts; accumulative coasts; coastal erosion; Gydan Peninsula

## 1. Introduction

The coasts of the Arctic seas are retreating at an average rate of 0.5 m/yr [1]. Spatial variability of coastal erosion, both regional and local, is determined by the structure of the coast (the coastline configuration, the width of the beach, the coastal bluff height, permafrost features, ice content, etc.) and hydrometeorological conditions (the temperatures of air and water, wave parameters, the length of the ice-free period, the frequency and the intensity of storms, etc.). An increase in coast retreat rates has been observed in many regions of the Arctic in recent decades; researchers often attribute it to climate change [2–5]. Climate changes, primarily sea level rise, will lead to increased erosion of seacoasts worldwide [6]. According to recent forecasts [7], by the end of the 21st century, the average rate of erosion of the Arctic coasts will increase and will probably exceed its historical range of variability in a wide range of climate scenarios. The sensitivity of thermal abrasion to warming will approximately double, reaching 0.4–0.8 m per year per °C by the end of the century [7]. Researchers predict that the average coastal erosion rate for the Arctic will increase from 0.9 ± 0.4 m/year in 1850–1950 to 1.6 ± 0.5, 2.0 ± 0.7, and 2.6 ± 0.8 m/yr by the end of the 21st century (2081–2100) under anthropogenic climate change according to the SSP1-2.6, SSP2-4.5, and SSP5-8.5 scenarios, respectively [7]. Despite the consensus in predicting the increased erosion of the Arctic coasts, the accuracy of forecasts is limited by insufficient initial data. Data on the dynamics of the

Arctic coasts currently remain fragmentary due to the insufficient spatial and temporal resolution of the available initial data [1,8,9].

Most of the studies of Arctic coasts are focused on the dynamics of thermo-abrasional coasts [3,9–12]. This type of coast has a pronounced coastal cliff, in which erosion and destruction lead to coastal retreat. However, in recent decades, due to global climate change and warming in the Arctic region, the coasts that were previously stable or advancing also began to erode. These are the first coasts formed during the Holocene due to marine sedimentation. Signs of ongoing erosion are often not expressed in their morphology due to the absence of coastal cliffs, but an analysis of multi-temporal aerospace and field data showed that such coasts are also subjected to active destruction. Many research studies have been dedicated to understanding the dynamics of accumulative sandy coasts worldwide [13–18]; however, few of them have studied this type of coast in permafrost conditions [9,19,20]. Accumulative low-lying coasts are highly sensitive to oil and oil product spills [21,22]; thus, the study of their dynamics is of high importance for the industry.

Human activity on the Arctic coast is constantly expanding. From 2000 to 2020, the area of built-up territories within the Arctic coastal zone increased by 15% [23], with the main increase observed in Russia. For Arctic settlements, the risks of coastal retreat caused by climate change will increase in the 21st century [24]. Within the settlements, it is often difficult to separate the natural factors of coastal dynamics activation from the anthropogenic ones.

The studied key site is a low sandy accumulative coast, where construction has been carried out since 2014. This study aims to assess the rates of retreat/advance of the coastline before and after construction based on the analysis of field and remotely sensed data and to determine the main drivers of the coastal processes during different time periods.

## 2. Study Area

The studied coast is located on the Gydan (eastern) coast of the Ob Bay of the Kara Sea (Figure 1). The Ob Bay, or the Gulf of Ob, is a unique estuary more than 800 km long formed by the Ob River. It is one of the world's longest estuaries. The gulf is elongated from north to south and is relatively narrow (about 60 km wide) and shallow (from 25–35 m deep in the north to 5–10 m deep in the south).

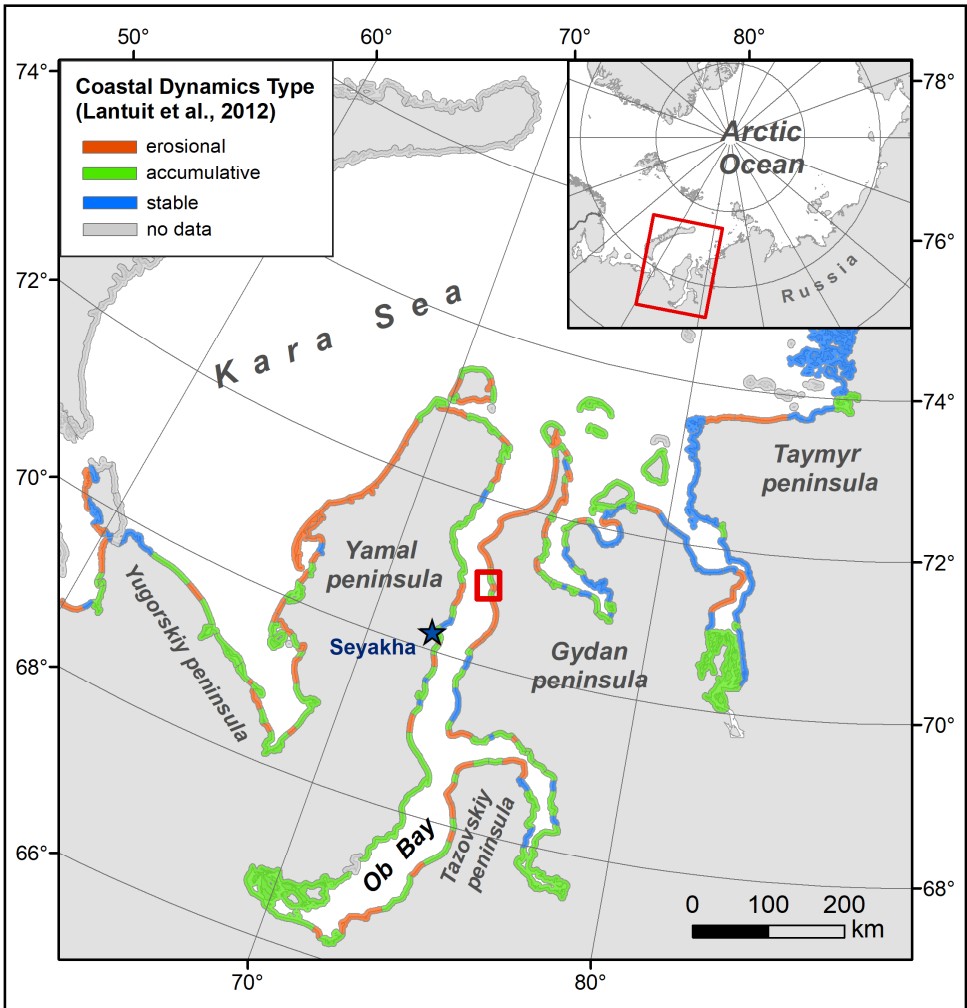

**Figure 1.** Study area. Red square shows the location of the study area; blue star shows the location of Seyakha weather station; Coastal Dynamics Types – from [1].

The coastal dynamics of the Gulf of Ob are generally determined by the complex interaction of marine, estuary, and river factors. The intensity of the marine impact, primarily tides and sea waves, decreases from north to south. On the contrary, the discharge current of the Ob River weakens from south to north. According to the prevailing factors of coastal dynamics, the Gulf of Ob is divided into three regions: southern (with a predominance of river runoff), middle (intermediate), and northern (with a predominance of marine impact and estuary processes) [25].

A 7 km section of the eastern coast has been studied in the middle part of the gulf between the mouths of the Khaltsanayakha River in the northwest and the Nadyaipingche River in the southeast. Here, the depth of the Gulf of Ob reaches its maximum [26] of 21 m at a distance of 10–15 km from the coast. Surface currents are directed from south to north along the shallow eastern coast of the bay and from north to south along the shallow western coast [27]. Based on geomorphological data [25], a local zone of divergence of alongshore sediment fluxes was identified in the studied section of the eastern coast. This zone creates a high risk of the onset/intensification of coastal erosion under changing natural and anthropogenic conditions.

This low-lying flat sandy coast—the so-called "laida" (low Holocene accumulative marine terrace inundable during the highest storms and surges)—is typical for the Ob Bay and other gulfs of the southwestern Kara Sea. Despite being accumulative in the recent past and still having a morphology of typical accumulative coasts, such coasts of the Kara Sea gulfs may retreat at a rate of 0.2–0.7 m/yr [28].

In the studied area, the laida is 0.5 km wide and 1–2 m above sea level. The surface comprises a series of sandy beach ridges up to 30 m wide and up to 1 m high, separated by swales filled with shallow (up to 1 m deep) lakes or swamps. Further inland, the laida is replaced by a marine terrace of 30–45 m height [29]. The underwater nearshore slope is shallow—0.1–0.5°. Tidal flats along the coasts are 70–100 m wide, and the beach has a width of 4–15 m. Thus, the coast is an accumulative type, i.e., it was formed due to marine sediments' accretion in the coastal zone and the advancing of the coastline.

The coast comprises sands (small grains 0.05–2 mm in diameter with silts 0.02–0.05 mm in diameter) interbedded with peat and detritus of alluvial–marine origin. Saline frozen soils with a temperature of −5 °C are widespread in the study area under the laida surface onshore, and they provoke thermoerosion, thermokarst, frost cracking, solifluction, and frost heaving [30].

The rivers on the borders of the studied coast serve as an additional sediment source for the coastal zone. The 0.9 km long beach ridge south of the mouth of the Khaltsanayakha River is currently subject to erosion. From 1972 to the present, no spits have been formed at the river mouths. At the given intensity of longshore sediment transport, the material brought by the rivers is insufficient for the formation of accumulative forms. The Khaltsanayakha River limits the study area from the northwest. The 55 km long meandering river has a catchment area of 210 km²; its banks are mainly composed of sandy soils. The Nadyaipingche River, which bounds the studied coast from the southeast, is much smaller—its length is 20 km, and the catchment area is 65 km².

The climate of the region is polar marine. The average annual temperature is about -10 °C (the average air annual temperature at the closest meteorology station, Seyakha, according to [31]). In winters, winds blow from the land to the sea and have a predominant southern direction; in summers, winds move in the opposite direction. Wind rates do not vary significantly; the annual amplitude does not exceed 1–3 m/s. The mean wind rates are highest in autumns and winters (up to 7–8 m/s). The range of water-level fluctuations in the Gulf of Ob in this area is up to 1.3 m, depending on tides and surges. The maximum calculated annual sea level of repeatability, 1 time in 50 years, exceeds the average level by about 1.5 m, which means that in this case, the water can cover a huge part of the low-lying marine terrace up to 2 m in height. This part of the Kara Sea is covered by ice for 7–8 months per year, from November to June–July [32]. In recent decades, there has been a trend of rising summer temperatures, prolonged ice-free period duration in the sea [32], and an increase in storm frequency due to global climate change [33].

## 3. Materials and Methods

### 3.1. Remote Sensing Data

An analysis of the coastline dynamics was carried out using satellite images of high spatial resolution with different time series. This is a traditional method used in coastal dynamics studies [2,5,34–36]. The position of the coastline on the erosional retreating coast is usually clearly detected as a cliff-top line. For accumulative coasts with no evident scarp (cliff/bluff), formed by waves, demarcating the coastal line is more challenging, as a coastal line proxy may be considered [37,38]:

1. Shoreline property—sea—land border. As a rule, this is easy to identify in an image; nevertheless, its position on accumulative low-lying coasts varies significantly (up to the first tens of meters laterally) during the season and even during the day, depending on the run-up of individual swell or wind waves and due to tidal, surge, and storm fluctuations of the sea level;
2. High water line or mean water line, or wet–dry line—approximated and averaged shoreline. This varies significantly from one day to the next depending on wind and wave conditions, tides, surges, and is a product of short-term beach erosion/accretion episodes, etc. It can be detectable in the field or with the use of DEM;

3.  Vegetation line—border of dense vegetation cover. This is the upper boundary of the coastal zone (similar to the cliff edge at the erosional coasts)—the upper boundary of the wave run-up, averaged over recent years. Thus, this boundary is more stable than the coastline, and its movement can better represent coastal dynamics. However, its position is difficult to trace on satellite imagery due to wind-blown sand from the beach deposited on the laida, causing the blurring of this border. The dense vegetation boundary is also highly sensitive to human-made activities;

4.  Scarp (cliff/bluff) top or baseline, if there is one.

The shoreline position on accumulative coasts varies significantly, even during the day; in this study, to characterize lithodynamic processes, besides a shoreline proxy (the boundary between the water's surface and land—Figure 2a), we considered a vegetation line—a dense vegetation boundary (Figure 2b). According to the results of our calculations for the studied section of the coast, the shoreline and the vegetation boundary movement show similar trends. The vegetation line proxy is more stable and reliable for long-term dynamics analyses; however, for our area of study, this line was complicated to trace due to the sandy aeolian cover widely distributed on the laida surface (Figure 3a). Moreover, with the construction of facilities, the vegetation line was almost completely destroyed in most of the territory (Figure 3b). Thereby, in this paper, we mainly used the shoreline as a proxy of the coastal line's position.

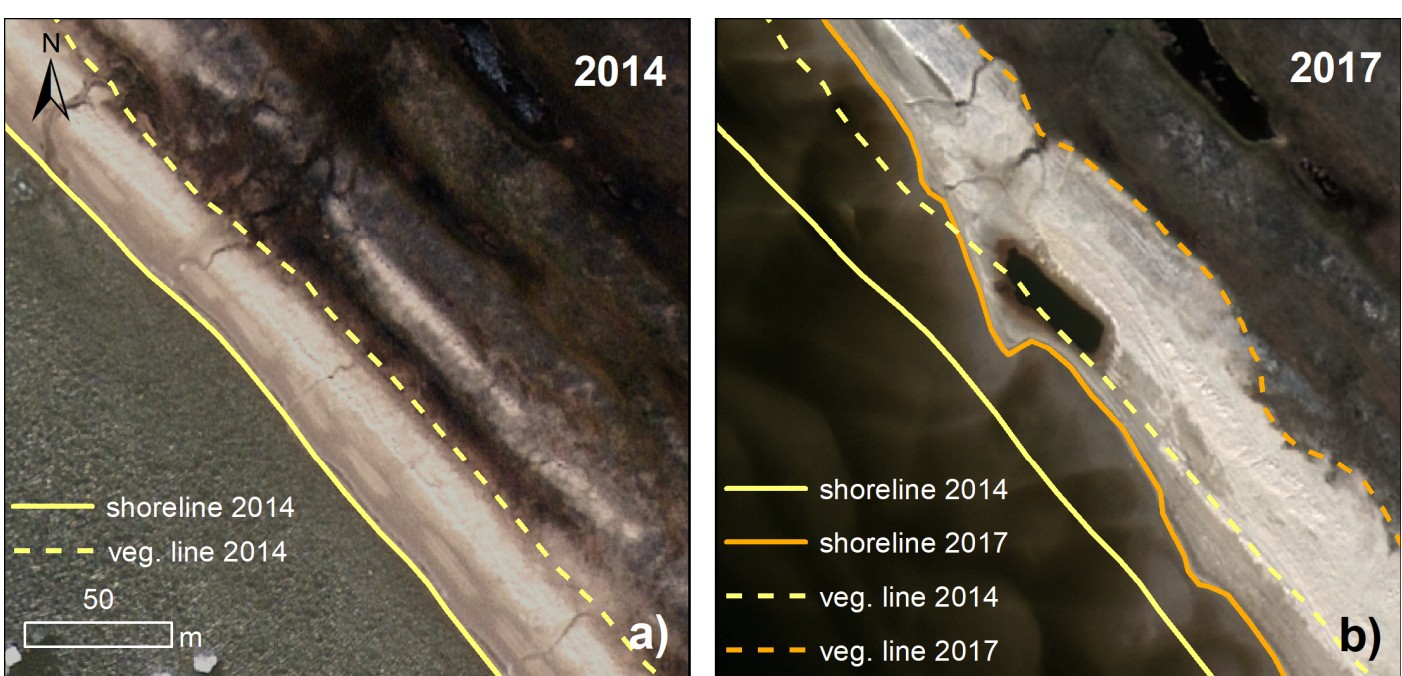

**Figure 2.** Coastal lines on the satellite images: (**a**) WV-2 2014 and (**b**) WV-2 2017.

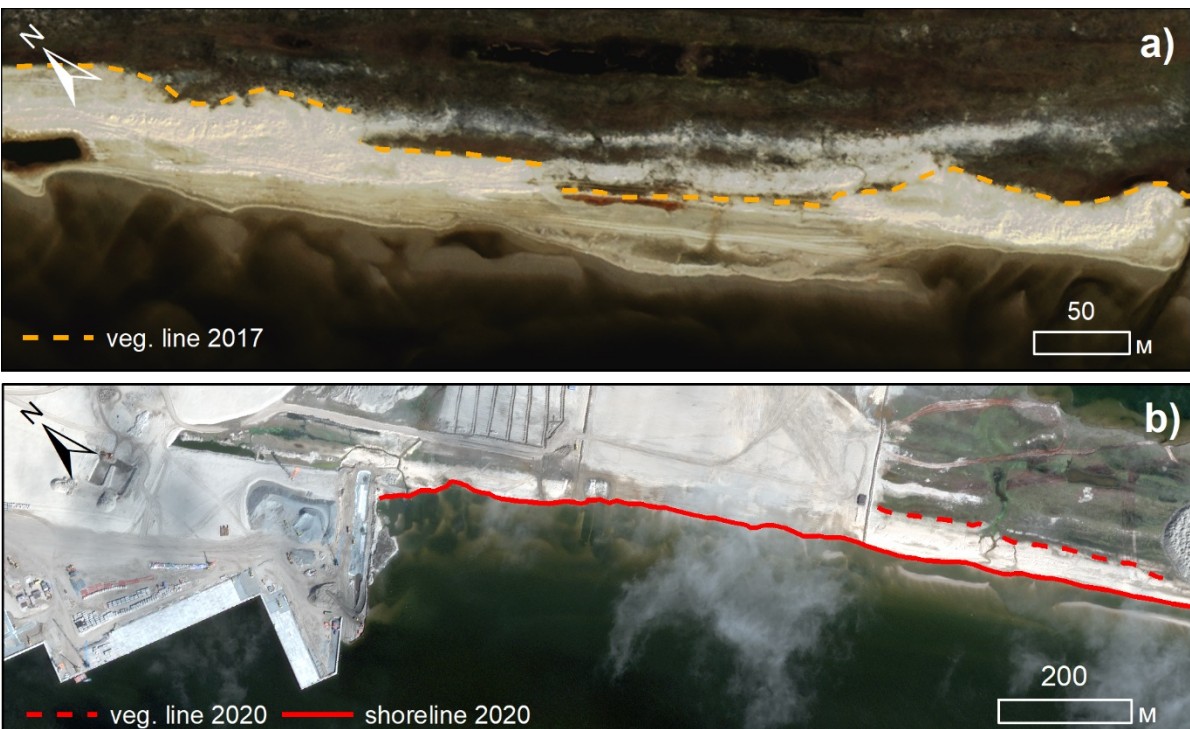

**Figure 3.** Challenging tracing of coastal lines position: (**a**) due to sandy aeolian cover on the laida surface (WV-2 2012); (**b**) due to construction in the coastal zone (WV-2 2020).

We used space images WorldView-2 and GeoEye-1, purchased from Maxar Technologies, with a spatial resolution of up to 0.5 m (panchromatic channel), taken on 23 August 2012, 29 June 2014, and 13 July 2014 (mosaic), as well as 14 July 2017 and 24 July 2020. All images were processed by a pan-sharpening operation to improve the spatial resolution of multispectral images. To improve their georeferencing, the images were orthorectified using freely distributed DEM ArcticDEM R.7 [29] with a 10 m resolution in the WGS84 coordinate system, UTM zone 43N. To improve the accuracy of the results, images were manually referenced using at least 10 points for each set of images in ArcGIS using 2nd-order Polynomial function for the 2014 and 2020 images and a 3rd-order Polynomial function for the 2012 image (all linked to the 2017 image). The final image referencing accuracy amounted to the following: for images of 2017 and 2020—0.4 m; for images of 2014 and 2017—0.5 m; for images of 2012 and 2017—1.0 m.

Images were processed and interpreted using ArcGIS 10.5 software [39]. The coastline was digitized at a scale of 1:500. The calculation of the rates of coastline movement was carried out using the ArcGIS Digital Shoreline Analysis System (DSAS) 5.0 extension [40]. This program automatically builds transects normal to the general direction of the coastline (baseline) with an optional spacing along the baseline (50 m in our case). A date is assigned for each coastline. Thus, for each transect, the rate for a specific time period is calculated by dividing the distance between the coastlines by that time period.

Considering the temporal variability of coastline movement rates, we should take into account that the first time interval (1972–2012) is much longer than the other three (2012–2014, 2014–2017, 2017–2020), so it is not appropriate to compare the average values of these periods. Nevertheless, this time span is crucial for the temporal analysis of coastal change as it represents the coast's natural state before the start of technogenic transformations and shows a general view of the dynamics of the studied coast during this period.

### 3.2. Analysis of Factors Influencing Coastal Dynamics

To understand the contribution of different factors to the dynamics of the studied coast and reveal a possible technogenic impact on these dynamics, we first assessed so-

called hydrometeorological forcing (HMF) [12,33,41]. This parameter is also called hydrometeorological stress/potential and reflects the combined effect of the changes in the principal hydrometeorological conditions on the coastal dynamics. For the Arctic coasts, it usually includes two main factors enhancing each other: thermal and wind-wave energy. Thermal energy (TE), also called thermal potential [41], depends on the positive air temperatures, driving the thawing of permafrost in coastal bluffs. The wind-wave energy (WWE), or wind-wave potential [41], is determined by the ice-free period's duration, length of the wave fetch, sea depth, and wind velocity; it is responsible for the erosion of coastal bluffs and removal of thawed material by waves.

Thermal energy was estimated by calculating the air-thawing index, which showed the sum of daily average temperatures above 0 °C per year. To calculate the wind-wave energy flux, we applied the Popov–Sovershaev method, which is based on the dependence of the energy flux on the ice-free period's duration, wave fetch along the wave-dangerous wind direction, wave-dangerous wind direction frequency, and wind speed in the third degree [33].

We obtained data on temperature, wind direction, wind speed, and wave-dangerous wind direction frequency for the closest available meteorology station, Seyakha, from [42] (Figure 1). Unfortunately, the available archive only contained the necessary data for the period beginning in 2010, which is why we do not provide the hydrometeorological forcing assessment for the first time period, 1972–2012. The ice-free period duration was estimated by the data on sea-ice distribution from [43]. Wave fetch and wave-dangerous wind directions were derived from the ETOPO-1 digital elevation model [44].

Human activity has influenced the coastal dynamics of the studied section since the beginning of port facilities' construction in 2014. The details of the project's construction (including the direct impact on the coastal zone) are restricted corporate data. Accordingly, in this article, we can only partly analyze the role of technogenic factors based on open data, such as satellite imagery. For example, the expansion of the built-up area in the central part of the studied coast can be seen in the images. However, the impact of construction is not limited to the areas of artificial coasts since the dynamics of coastal segments without constructions are also affected by the removal of sands from the beach and tide flat or by the dredging inevitable during port construction.

## 4. Results

### 4.1. Coastal Dynamics

Considering the dynamics of the coast for different time periods, we can observe its substantial variation with time.

#### 4.1.1. Before the Beginning of Construction

Despite its accumulative morphology and origin, the coast was slightly retreating (−1.1 m/yr on average for the shoreline (Table 1, Figure 4a) and −0.1 m/yr on average for the vegetation line (Table 2, Figure 4b)) during the whole period of study (1972–2021). The greatest retreat of the shoreline took place in the central part of the studied area around the constructed port (up to 2.5 m/yr—Figure 5) and the northwestern part (up to 2.3 m/yr). In the southeastern half of the area, the shoreline retreat rates were less than 1 m/yr. The vegetation line retreated the most (up to −0.9 m/yr) to the northwest of the pier (Figure 6), following the shoreline retreat and due to heavy vehicle passage. We suppose that construction in the central part of the study area is the main reason for the spatial variability of coastal erosion since the coast receded relatively evenly over the entire 7 km before 2014. Most of the study area had been built up by 2020, so it was impossible to calculate the retreat for the 1972–2020 time period for a huge part of the coast (23% of the shoreline and 39% of the vegetation line). However, based on the dynamics of the segments without constructions, the general tendency to retreat can be clearly seen.

**Table 1.** Dynamics of the shoreline.

| Time Period | Rate of Retreat/Advance, m/yr | | | Dynamics, % of All Shoreline | | |
|---|---|---|---|---|---|---|
| | Average | Max | Min | Retreat | Advance | No Data |
| **All periods of study** 1972–2020 | **−1.1** | **0.1** | **−2.5** | **75** | **1** | **23** |
| **Before the beginning of construction** 1972–2014 | **−0.6** | **0.2** | **−2.4** | **92** | **5** | **2** |
| 1972–2012 | −0.8 | −0.03 | −2.5 | 94 | 0 | 6 |
| 2012–2014 | 4.1 | 21.6 | −3.4 | 17 | 77 | 6 |
| **After the beginning of construction** 2014–2020 | **−4.9** | **2.1** | **−16.1** | **69** | **7** | **24** |
| 2014–2017 | −5.8 | 5.8 | −28.5 | 77 | 16 | 7 |
| 2017–2020 | −3.8 | 3.9 | −13.9 | 65 | 10 | 25 |

**Table 2.** Dynamics of the vegetation line.

| Time Period | Rate of Retreat/Advance, m/yr | | | Dynamics, % of All Shoreline | | |
|---|---|---|---|---|---|---|
| | Average | Max | Min | Retreat | Advance | No Data |
| **All periods of study** 1972–2020 | **−0.1** | **0.6** | **−0.9** | **19** | **11** | **39** |
| **Before the beginning of construction** 1972–2014 | **0.9** | **1.6** | **0.0** | **2** | **80** | **18** |
| 1972–2012 | 0.9 | 1.7 | −0.2 | 2 | 78 | 20 |
| 2012–2014 | 2.1 | 15.5 | −4.7 | 29 | 34 | 6 |
| **After the beginning of construction** 2014–2020 | **−3.4** | **1.3** | **−8.6** | **35** | **9** | **57** |
| 2014–2017 | −5.2 | 2.9 | −28.8 | 67 | 22 | 11 |
| 2017–2020 | −2.1 | 9.1 | −10.9 | 33 | 14 | 53 |

During the period before the construction (1972–2014), the studied coast was quite stable, with a slightly retreating shoreline (−0.6 m/yr on average—Table 1, Figure 7) and slightly advancing vegetation line (0.9 m/yr on average—Table 2, Figure 8). Almost the entire (92%) shoreline retreated, and the maximal retreat rate (2.4 m/yr) occurred in the far NW of the area close to the mouth of the Khaltsanayakha River. Most of the vegetation line (80%) advanced; the greatest advance (1.6 m/yr) was described for the central part of the area.

**1972–2012.** During the first study period, the coastal dynamics were relatively slow: the shoreline was retreating (94% of it) with an average rate of −0.8 m/yr, and the vegetation line was advancing (78% of it) with an average rate of 0.9 m/yr. The shoreline retreated with the highest rate in the NW area around the Khaltsanayakha River mouth (up to −2.5 m/yr) and the SE area around the Nadyaipingche River mouth (−2.0 m/yr). The maximal advance of the vegetation line (up to 1.7 m/yr) was noted in the central part of the area. The rates of the shoreline and vegetation line movements did not change much along the coast.

**2012–2014.** The years of 2012–2014 was a period of relatively stable dynamics with dominating accumulation (advancing of the shoreline and vegetation line). About 3/4 (77%) of the shoreline and about 1/3 (34%, which is the greater part) of the vegetation line shifted seawards. The shoreline advanced at an average rate of 4.1 m/yr and a maximal rate of 21.6 m/yr in the SE area close to the Nadyaipingche River mouth. In the central part of the area, there was a retreat of the shoreline (17% of the coast) at rates up to −3.4 m/yr. The average rate of the vegetation line's movement was 2.1 m/yr; the maximal retreat (up

to –4.7 m/yr) took place in the central part of the area, and the maximal advance (more than 10 m/yr, up to 15.5 m/yr) took place in the SE part close to the Nadyaipingche mouth.

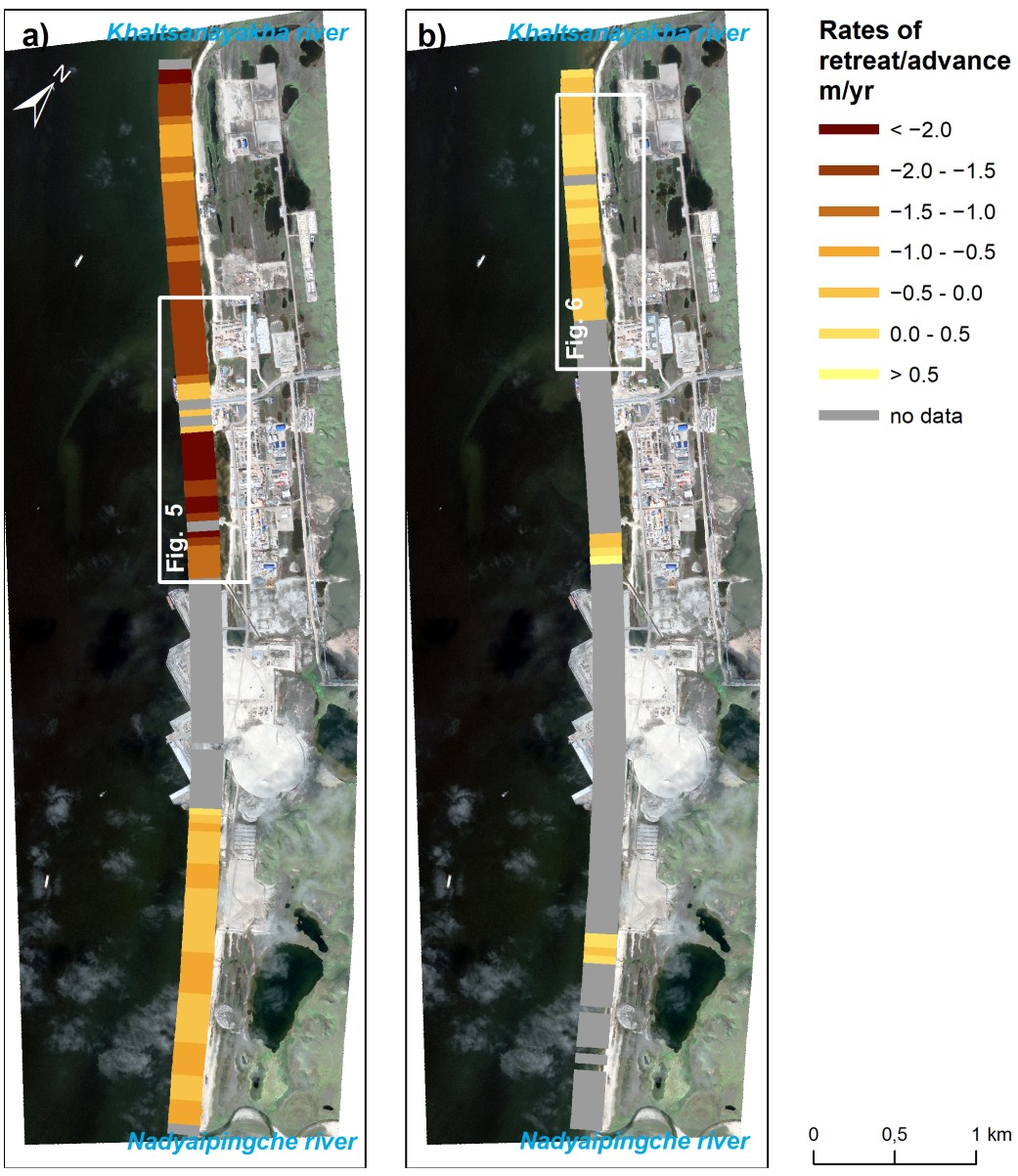

**Figure 4.** Shoreline (**a**) and vegetation line (**b**) dynamics for the whole period of study (1972–2020).

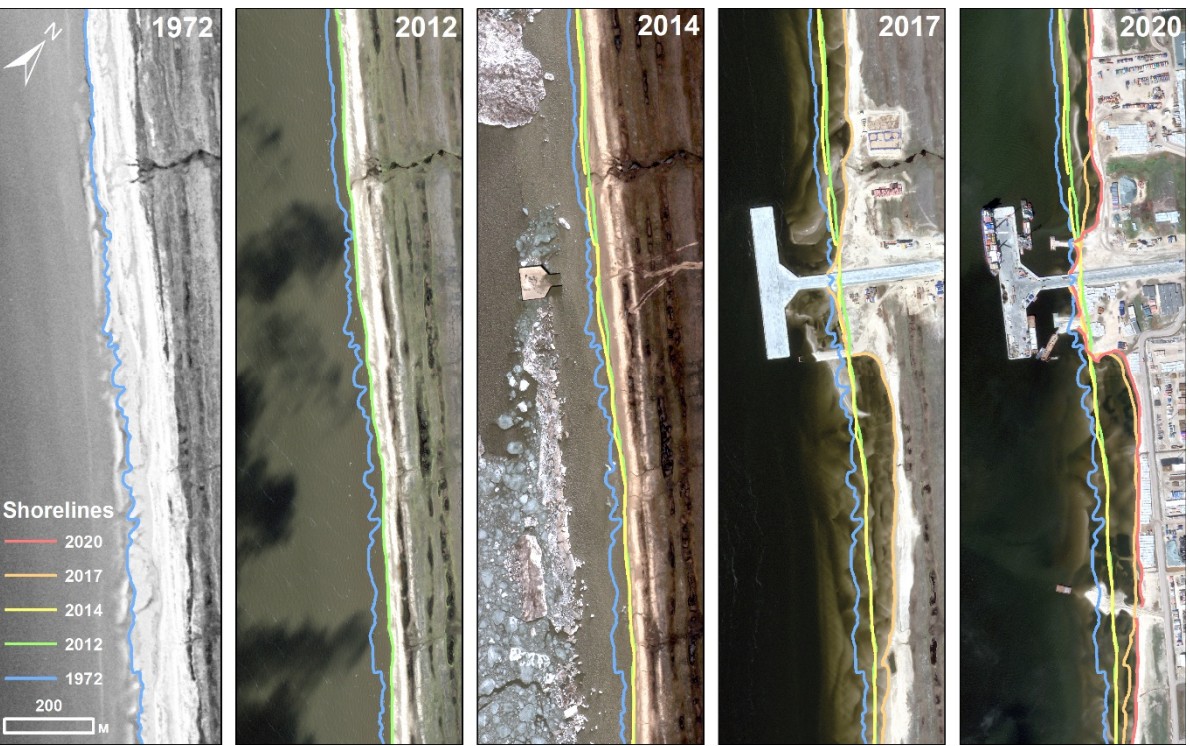

**Figure 5.** Shoreline position changes from 1972 to 2020 in the area with the highest rates of retreat.

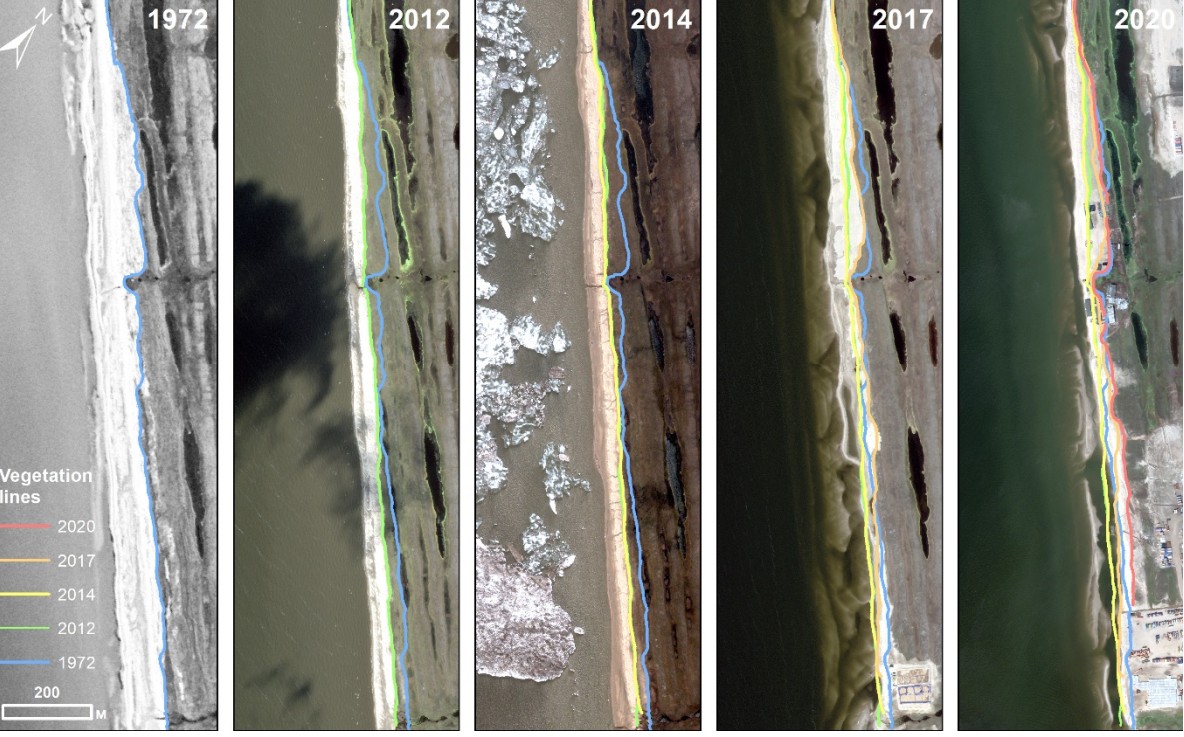

**Figure 6.** Vegetation line position changes from 1972 to 2020 in the area with the highest rates of retreat.

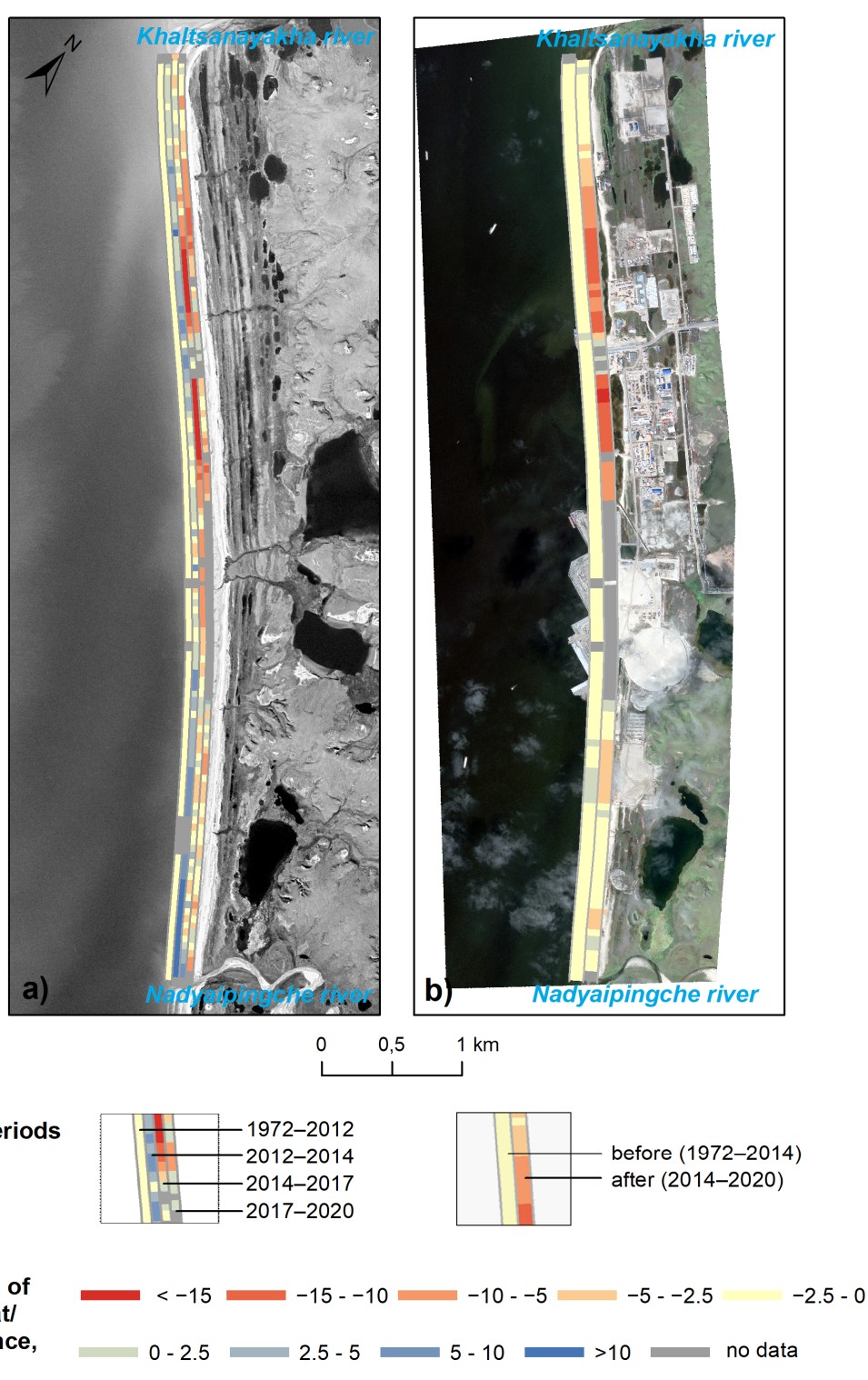

**Figure 7.** Shoreline dynamics (**a**) by different periods (1972–2012, 2012–2014, 2014–2017, 2017–2020), (**b**) before (1972–2012) and after (2012–2020) the beginning of construction.

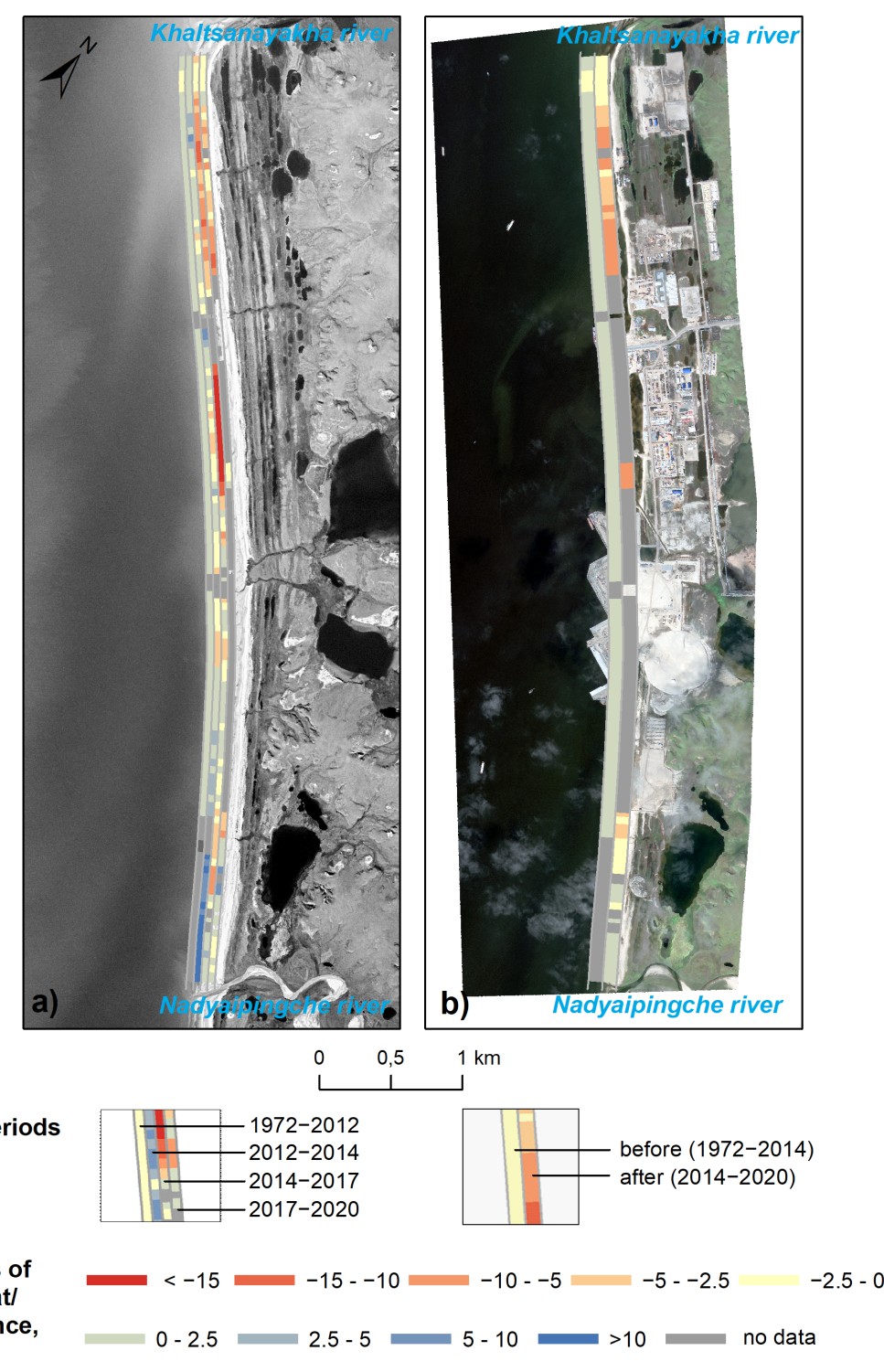

**Figure 8.** Vegetation line dynamics (**a**) by different periods (1972–2012, 2012–2014, 2014–2017, 2017–2020), (**b**) before (1972–2012) and after (2012–2020) the beginning of construction.

### 4.1.2. After the Beginning of Construction

After 2014, the dynamics of the coast changed substantially. In 2014–2020, most of the coast was retreating: 69% of the shoreline and 35% of the vegetation line (given the fact that a considerable part of the coast was built up by the gas project constructions and rates of coastal retreat/advance were not calculated here) (Tables 1 and 2, Figures 7 and 8). The average rate of shoreline movement in 2014–2020 amounted to –4.9 m/yr (that is, eight times higher than the average rate in 1972–2014). The average rate of vegetation line

movement in 2014–2020 amounted to –3.4 m/yr (whereas in 1972–2014, the coast was advancing in general, and the average rate was 0.9 m/yr). Erosion of the coast (based on the changes of both the shoreline and the vegetation line) was the most considerable in the central part of the studied area between the constructed pier and terminal: it amounted up to –16.1 m/yr for the shoreline and up to –8.6 m/yr for the vegetation line.

**2014–2017.** The years of 2014–2017 was a period of sharp intensification of erosion (expressed in both the shoreline and the vegetation line proxies), and it coincided with the initial stage of the terminal construction. The situation was the opposite of the previous period—erosion was observed on about 3/4 of the coast (77% of the shoreline and 67% of the vegetation line). The shoreline retreat rate averaged –5.8 m/yr for the vegetation line to –5.2 m/yr. Directions and values of coastal movements varied substantially along the coast. The most intensively retreating section was about a 3 km long coastal segment in the middle part of the study area on both sides of the constructed pier (Figure 5). Erosion rates here were more than –15 m/yr, up to 28.5 m/yr for the shoreline, and up to –28.8 m/yr for the vegetation line. In the SE part of the area closer to the Nadyaipingche mouth, both the shoreline and the vegetation line experienced seaward movement at the highest rate of 5.8 m/yr for the shoreline and 2.9 m/yr for the vegetation line.

Within this area, traces of sediment removal from the beach and the tide flat are visible in the space images (Figures 2b and 3a). During this period, to the northwest of the constructed terminal, the chain of longshore bars formed in the nearshore zone at the site of the 2014 coastline.

**2017–2020.** The last time span, 2017–2020, was a period of continued strong erosion but with lower retreat rates (1.5–2 times lower than the previous period). Erosion was observed on about 2/3 of the shoreline; the average of all the shoreline rates amounted to –3.8 m/yr. Most of the vegetation line was undisturbed by construction and available for monitoring. It was retreating (33%), and the average rate of its movement amounted to -2.1 m/yr. The section with the most intensive erosion moved from the central part of the coast to the NW, and now, it is a 0.5 km long segment of coast ~2 km to the NW from the pier (Figure 6). The shoreline moved landward here at a rate of up to –13.8 m/yr, and the vegetation line moved at a rate of up to –10.9 m/yr.

Thus, before the development of the studied area (1972–2014), the coast dynamics were low, with a slight tendency to retreat. With the beginning of construction, the coast type changed to active erosional. The maximum rates of retreat were observed during 2014–2017, which corresponded to the period of the most intensive technogenic changes in the coastal zone. Spatially, the greatest coastal erosion occurred in the construction area, and the rate of retreat increased in some places. Relatively stable coasts of the Gulf of Ob with similar rates of progradation/retreat observed in recent decades differ in resistance to technogenic impact and climate changes. For example, the construction of the Sabetta port on the opposite side of the Gulf of Ob, the eastern coast of Yamal Peninsula 65 km away from the study site, has not led to such a sharp increase in retreat rates. We suppose that this is due to the difference in natural conditions, namely, nearshore bathymetry and longshore sediment transport.

*4.2. Hydrometeorological Forcing*

Calculated values of wind-wave (Figure 9) and thermal (Figure 10) energy for the distinct time periods from 2011 to 2020 varied significantly; however, they showed a slight but clear trend of increasing. Maximal wind-wave forcing affected the coast in 2016, 2012, and 2018, and minimal forcing occurred in 2013 and 2015. The maximal thermal effect occurred in 2016 and 2020, and the minimal effect occurred in 2014. The combined average hydrometeorological forcing was maximal in the 2014–2017 period; it was minimal in 2012–2014.

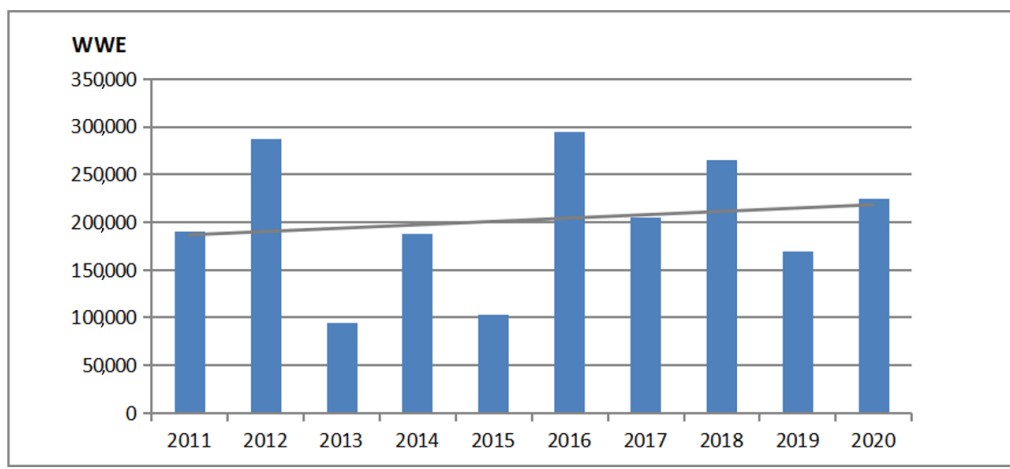

**Figure 9.** Wind-wave energy (WWE, kg·10³ per ice-free period) change from 2011 to 2020.

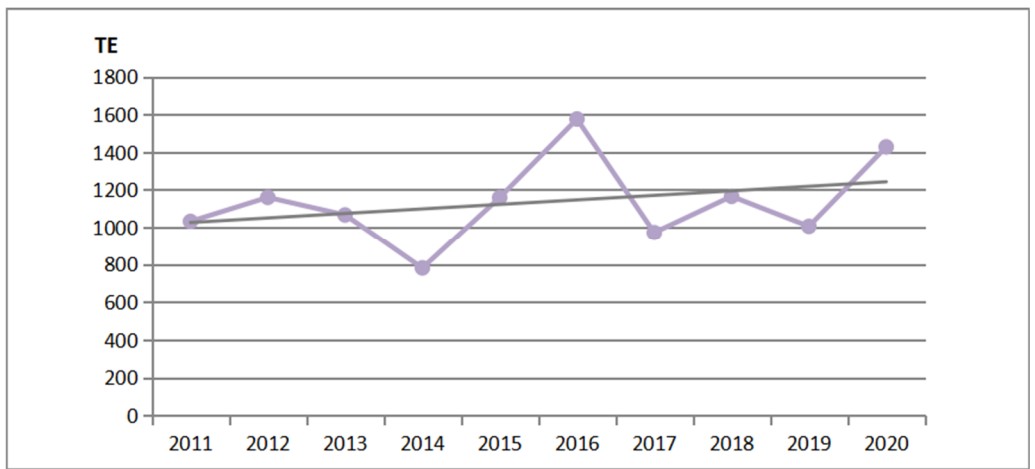

**Figure 10.** Thermal energy (TE, sum of daily average temperatures above 0 °C per year) change from 2011 to 2020.

## 5. Discussions

### 5.1. Factors of Spatial Variation of Coastal Dynamics

The spatial variability of coastal dynamics (change in the rates of coastal retreat/advance along the coast) is determined by such factors as the geomorphological structure (aspects of the coast, height of the cliff, width of the beach, and other parameters of coastal landforms), lithological features (sediment composition: petrography, grain size, and other properties, sediment balance and longshore sediment fluxes in the nearshore zone), and parameters of permafrost (ground ice content).

Long-term dynamics of the studied shoreline (Table 1, Figure 4a) and vegetation line (Table 2, Figure 4b) do not show substantial variations along the coast: rates of movement for the shoreline vary from −2.4 to 0.1 m/yr, and for the vegetation line, they vary from -0.9 to 0.6 m/yr. This is due to the homogeneous lithological and geomorphological composition of the coast. The deviations in values are the highest in the construction area in the central segment of the coast.

In the period before the construction (1972–2014), spatial variations of the rates of coastal changes were minimal and did not exceed 2.6 m/yr for the shoreline (Figure 7) and 1.6 m/yr for the vegetation line (Figure 8) (whereas in 2014–2020, the differences were 18.2 and 9.9 m/yr, respectively), although the first period was much longer and it was not correct enough in its data to compare its average values. However, looking at the short period before construction, 2012–2014, we found the areas of predominate accumulation (progradation of the coast): first is the SE quarter of the studied coast, especially the part at the

Nadyaipingche River mouth (up to 21.6 m/yr); some sections in the NW part of the area in front of the mouths of small creeks also experienced predominate accumulation.

With the start of construction, the coast dynamics changed considerably: erosion of the coast was activated, and spatial variations of rates grew. The greatest retreat of the coast took place in close proximity to the constructed terminal (up to about –29 m/yr in 2014–2017). As we moved away from the terminal to the SE, the coastal retreat rate diminished, and the coast became more or less stable or even slightly advancing (near the river mouths)—the maximum advance was about 6 m/yr. To the NW, retreat of the coast was slighter but still quite considerable (from –1.0 to –2.3 m/yr). Intensive retreat around the terminal is most probably related to the technogenic impact on the coast during the terminal's construction: for example, due to the vegetation cover disturbance as a result of vehicles pressing on the ground, due to the sediment disturbance and excavation from the beach and sea bottom for construction, due to disturbance of the sediment flow after the terminal construction, and other causes (Figure 5).

The advance of the shoreline (m/yr) in the southern part of the coast near the temporal port constructed in 2018–2019 may be related to anthropogenic filling of the sandy banks on the beach or filling of the angle directed toward the longshore flux by sediments.

According to instrumental observations, the longshore sediment flow in the studied area had two prevailed directions: SE and NW. The constructed long terminals apparently blocked the flow and entrapped a large portion of moving sediments, which resulted in the enhanced accumulation of sediments on both sides of the construction area and enhanced erosion in the area near the terminal (Figures 7 and 8). Moreover, according to the previous regional assessments of the Ob' Bay coastal dynamics [25], the zone of divergence of the longshore flows is located approximately in the area of our study. This may also have led to the enhanced erosion of this section of the coast.

### 5.2. Temporal Variation of Coastal Dynamics

The temporal variability of coastal dynamics mainly depends on the changes in hydrometeorological conditions from year to year. To compare these conditions for the different periods of study, we calculated the average values of wind-wave and thermal energy for these periods (Figure 11).

Changes in coastal dynamics over time are significant (Tables 1 and 2, Figures 5–8).

Before the terminal's construction, the coast was quite stable with slight transformations of separate sections of the coast (the average rate during the period of 1972–2014 for the shoreline was –0.6 m/yr, and for the vegetation line, it was –0.9 m/yr), and it makes sense that this coast was accumulative due to its morphology and origin. However, in the following years (2014–2020), the coast predominately experienced a negative sediment balance and retreating, with an average rate of –4.9 m/yr (–3.4 m/yr for the vegetation line). The coast retreated the most intensively in 2014–2017 (–5.8 m/yr for the shoreline), coinciding with the period of maximal hydrometeorological forcing (Figure 11). Additionally, this time was a period with the most intensive construction work.

Regarding the correlation of HMF with rates of coastal retreat/advance (Figure 11), we see that in general, maximal values of HMF correspond to maximal coastal erosion and vice versa, especially the values of WWE. TE's influence is weaker on this coast, since it is composed of low ice content deposits.

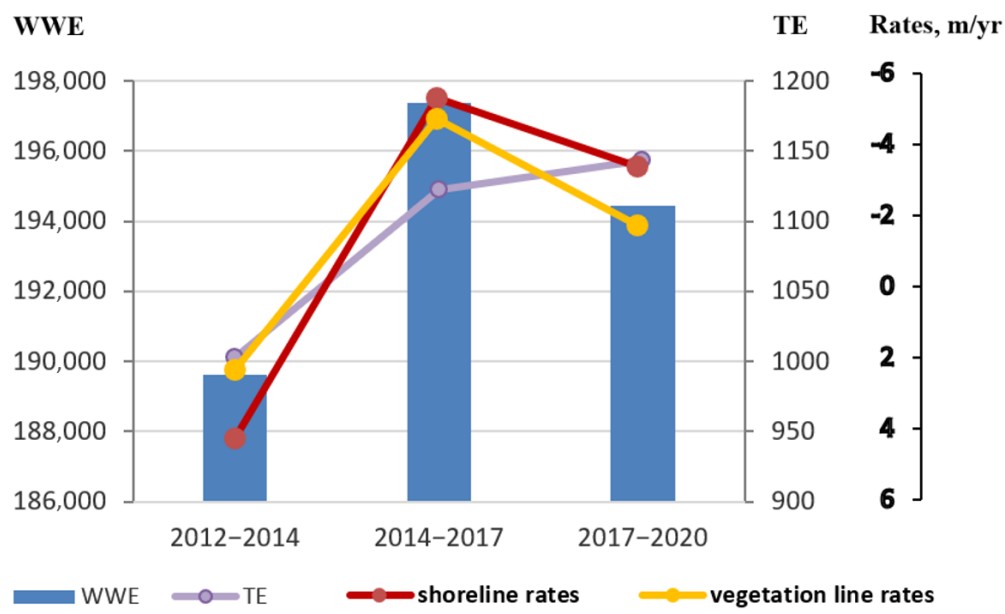

**Figure 11.** Hydrometeorological forcing and rates of coastal retreat/advance for the periods of study. Wind-wave energy (WWE, kg·10³ per ice-free period). Thermal energy (TE, sum of daily average temperatures above 0 °C per year).

However, we propose that such a dramatic intensification of coastal erosion as during 2014–2017 cannot only be explained by the enhanced HMF: the rates increased up to several tens meters, whereas the average rates of retreat of the Arctic coasts amount to -0.5 m/yr, and for the Kara Sea coasts, −0.7 m/yr [1]. Moreover, the suggestion of technogenic influence is supported by the fact that the greatest retreat was observed in close proximity to the constructed terminal (Figure 5).

In the last time period (2017–2020), the erosion rate decreased in the most intensively retreating area around the constructed plant, which is related to the diminution of WWE pressing but also may indicate a relative stabilization of the lithodynamic system's state after intensive impact at the previous stage.

The calculated ice-free period duration trend amounted to plus 10–12 days/decade (1979–2019) at the Gydan Peninsula coast at 71.5° N [32]; thereby, the role of hydrometeorological parameters in coastal dynamics of the study area is expected to increase in the upcoming decades. Climate change will lead to coastal retreat [45]. It is of great importance to assess the vulnerability of the studied coasts to the changing climate, as it was carried out for other regions of the world [46,47]. This will help develop an adaptation strategy of human activity to the hazards associated with sea level changes.

*5.3. Human Impact*

Despite the most intensive coastal erosion in the period of 2014–2017 correlating well to the increased hydrometeorological forcing at this period, we propose that the construction of the terminal impacted the coast and coastline movement, at least as soon as the most considerable changes were revealed directly close to the constructions, not along the whole area. We propose that significant local transformation of the coastal line, especially in 2014–2017 and later, was, to a great extent, related to the construction works. Over 6.5 km of the study area underwent an erosional process, with some areas retreating more or less than others. Human activity around the terminal may drive or enhance the collapse of the coasts by such processes as the disturbance of deposit flows along the coast and the disturbance of soils and vegetation cover by vehicles.

The construction has changed the sediment transport of Khaltsanayakha River. The 2020 image shows a temporary bridge 4 km upstream from the river mouth with wide

embankments on both banks of the river. Muddy river water indicates the partial erosion of this bridge. Without a special study, it is impossible to say whether the supply of material to the coastal zone has increased or decreased. On the one hand, the bridge could reduce the amount of sediments transported to the river mouth as well as the extraction of sand from the riverbed. On the other hand, the erosion of bridge embankments causes additional sediment supply.

In the coming decades, global climate warming will continue to enhance coastal erosion in the Arctic, according to most forecasts [45]. Technogenic pressure may continue to influence the coastal processes as well. In case of continued erosion, coastal monitoring may be needed. However, after the termination of construction, and in the case of reasonable environmental management, erosional processes may reduce and even give way to accumulation in the coming years, as it was before the start of development in this region.

*5.4. Comparison to Other Arctic Regions*

The retreat rates of the study area were considered in comparison to the average annual rates of coastal erosion in the Arctic. The mean annual rate of coastal retreat for the whole Arctic was about 0.57 m/y and 0.68 m/y in the Kara Sea [1]. The highest rate of coastal retreat was observed in the 2008–2009 period in Drew Point, Alaska, USA, amounting to 30 m/y [48], and for the Kara Sea, it was 3.3 m/y in 1989–1990 in Marre-Sale [49]; however, these rates were observed first of all for the erosional types of coasts.

In addition to the accumulative types of coasts in the Arctic, spits and barrier islands of the Beaufort seas have been studied. For example, in the southeastern Canadian Beaufort Sea between 1950 and 1985, the mean recession rate of the barrier islands was 3.1 m/yr, 1.7 m/yr for the spits, and 2.0 m/yr for the "detached spits" [19]. Along the Yukon coast, the highest rates of shoreline movement were observed along gravel beaches, barrier islands, and spits: they varied from −7.2 ± 0.2 m/yr to 5.3 ± 0.2 m/yr, and they showed an increase since the 1990s [4]. The coastline of the Elson Lagoon near Barrow, Alaska, retreated in 2002–2011, with an average rate of 2.17 m/yr, and it ranged from 1 to 4 m/yr [20]; for the historic period, it was 0.56–0.86 m/yr [50]. Within the Bering Land Bridge National Preserve, USA, shorelines associated with barrier islands have experienced the greatest variations, with mean rates of change up to −1.53 m/yr occurring between 2003 and 2014 along low-relief reaches of coast lacking foredunes in the supratidal zone [51].

## 6. Conclusions

The studied section of the Gulf of Ob coast was stable or accumulative until 2014 along all 7 km of the coast. In recent decades, enhanced hydrometeorological forcing and the construction of port facilities have led to considerable changes in the lithodynamic processes. In 2014–2017, the coastline retreated at an average rate of −5.8 m/ yr, and in 2017–2020, its average rate was −3.8 m/yr. The maximal rates of retreat took place near the new constructions. In the coming years, with Arctic climate warming and continued human impact, erosion of the studied coast may intensify.

The investigated eastern coast of the Gulf of Ob is more vulnerable to changes in both the natural environment (sea level rise, wave regime, etc.) and technogenic impact (construction in the coastal zone) compared to neighboring areas of technogenic development. Under changing climate, it is highly probable that coastal retreat will continue.

For engineering work at all stages from design to construction and operation, we recommend considering probable coastal retreat; identifying infrastructure that may be adversely impacted; determining primary factors driving active erosion (earthworks, deposits drift, etc.); and monitoring the coastal retreat to prepare potential mitigation measures for coastal infrastructure.

**Author Contributions:** Conceptualization, Y.S. and A.N.; supervision, Y.S. and S.O.; project administration, Y.S. and A.E.; methodology, A.N.; data acquisition, A.N. and N.B.; data processing, A.N.; visualization, A.N.; writing—original draft preparation, A.N. and N.B.; writing—review and

editing, Y.S. and N.B.; funding acquisition, Y.S., A.E. and S.O. All authors have read and agreed to the published version of the manuscript.

**Funding:** The research was funded by Russian Science Foundation, grant no 22-17-00097.

**Data Availability Statement:** Data available on request due to privacy restrictions.

**Acknowledgments:** Satellite images were partially provided by the Geoportal of Moscow State University. We acknowledge Helen Bideaud and Romain Jatiault for assisting in acquisition and processing of satellite images.

**Conflicts of Interest:** The authors declare no conflict of interest.

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
