# Peer review of "Dynamics of Low-Lying Sandy Coast of the Gydan Peninsula, Kara Sea, Russia, Based on Multi-Temporal Remote Sensing Data"

_remotesensing, doi:10.3390/rs15010048_

Round 1

Reviewer 1 Report

I appreciate the opportunity to review the article titled Dynamics of sandy low-lying coast of the Gydan Pen-2 insula, Kara Sea, based on multi-temporal remote 3 sensing data.  I also appreciate author’s ability to choose an insightful study, particularly one that focuses on coastal dynamics using geospatial techniques. After careful reading, I'd say the article has number of flaws which need to be fixed. Having said that, I have recommendations that I believe the authors will find useful and that, if implemented, would make this research suitable for publication in the journal.

#Title

Title of the article is somewhat unclear about the study area, country name should include in the title of the article

 # Abstract

 The given abstract requires significant change in terms of motivation and implication because it is not acceptable at its current stage. An abstract should begin with a general statement about the topic. Methods and possible outcomes are not expressed academically. The study's implication is unclear. At the end of the abstract, kindly include a statement outlining the study's implications. Do not use the term in this paper rather use in this study or research and delete lines 14 and 15

 # Introduction

 Introduction is written poorly without proper background and discussion of previous related studies and most importantly the research gap which helped to formulate this particular research. I do not see the idea of the article has been established properly. Without Scientific referencing and detailed analysis of those approaches, the conclusion and argument of the article do not sound appropriate. . I recommend the author to look at the latest scientific publication before jumping to any conclusion, as a lot might have been changed in the decade-long period.

In other way I would say in the Introduction, it is necessary to reconstruct the storyline of this manuscript. The introduction is the opening part of the scientific story to attract an audience and suggest the direction of your research. For this reason, you need to identify the problem that drives the research and introduce the key characters. If then, using the key characters, it is required to intertwine the scientific story concisely, systematically, and logically. Please rearrange the keywords so that the background of the study sounds appropriate.

 # Study Area

Provide a rationale why this area is selected for the study, too much of information need to shorten the study area section and I found plagiarized sentences too. Please change the study area map and highlight study area rather putting a square box in a regional settings.

 # Materials and method

 Entire methodology section is poor and vague which make the research unclear in terms of methodological design. It is hard to understand the direction of this research. It is necessary to reorganize the hierarchy of Materials and Methods such, overview of the method, Data Collection, and Data Analysis, validation etc. Please include a brief flowchart to help readers who are unfamiliar with this topic comprehend the study methodology. I do not see any flowchart for this study, please provide a brief but robust flowchart so that reader can be benefited from the study. Please make a lucrative chart using relevant graphics and symbol. Writing is not scientifically appropriate in this section. Please get some idea from similar articles online. Lines 133-213 are vague please shorten this section and only put relevant information’s. Figure 2 is irrelevant here.

 # Result and discussion

 Result sections could be improved at its present form it is hard to understand the finding of the study. As per as the discussion section is concern   This article does not have a proper scientific discussion section from where the reader can get comparative discussion on data and techniques relative to coastal dynamics. If possible please elaborate limitation, future direction and policy implication of this study

 # Conclusion

Should be revised, too much of generic sort of discussion here, I cannot see the brief of results of this study here. Please revise conclusion according. Supplementary document is highly desirable otherwise reader would not be benefited. I found 30 % similarity index for this article, I would recommend to authors to rewrite (report attached).

Author Response

Reviewer: I appreciate the opportunity to review the article titled Dynamics of sandy low-lying coast of the Gydan Pen-insula, Kara Sea, based on multi-temporal remote sensing data.  I also appreciate author’s ability to choose an insightful study, particularly one that focuses on coastal dynamics using geospatial techniques. After careful reading, I'd say the article has number of flaws which need to be fixed. Having said that, I have recommendations that I believe the authors will find useful and that, if implemented, would make this research suitable for publication in the journal.

The authors agree with the remarks of the reviewer. We tried to fix them all. Edits are commented below.

#Title

Reviewer: Title of the article is somewhat unclear about the study area, country name should include in the title of the article

Answer: country name included

 # Abstract

Reviewer: The given abstract requires significant change in terms of motivation and implication because it is not acceptable at its current stage. An abstract should begin with a general statement about the topic. Methods and possible outcomes are not expressed academically. The study's implication is unclear. At the end of the abstract, kindly include a statement outlining the study's implications. Do not use the term in this paper rather use in this study or research and delete lines 14 and 15

Answer: lines 14 and 15 were deleted.
added to the end of the abstract:
In the coming years, with Arctic climate warming, erosion of the studied coast will continue.

 # Introduction

Reviewer: Introduction is written poorly without proper background and discussion of previous related studies and most importantly the research gap which helped to formulate this particular research. I do not see the idea of the article has been established properly. Without Scientific referencing and detailed analysis of those approaches, the conclusion and argument of the article do not sound appropriate. I recommend the author to look at the latest scientific publication before jumping to any conclusion, as a lot might have been changed in the decade-long period.

In other way I would say in the Introduction, it is necessary to reconstruct the storyline of this manuscript. The introduction is the opening part of the scientific story to attract an audience and suggest the direction of your research. For this reason, you need to identify the problem that drives the research and introduce the key characters. If then, using the key characters, it is required to intertwine the scientific story concisely, systematically, and logically. Please rearrange the keywords so that the background of the study sounds appropriate.

Answer: Thank you, we fixed the introduction. We are quite sure we used all the latest data available on the topic.

 # Study Area

Reviewer: Provide a rationale why this area is selected for the study, too much of information need to shorten the study area section and I found plagiarized sentences too. Please change the study area map and highlight study area rather putting a square box in a regional settings.

Answer: We choose a section of the Gulf of Ob, where studies of coastal dynamics have never been carried out before. It should be noted that the results of detailed quantitative studies of the long-term dynamics of the shores of this bay of the Kara Sea have not been published before.

The study area map was partly corrected. The whole study area can be seen at figures with results.

 # Materials and method

Reviewer: Entire methodology section is poor and vague which make the research unclear in terms of methodological design. It is hard to understand the direction of this research. It is necessary to reorganize the hierarchy of Materials and Methods such, overview of the method, Data Collection, and Data Analysis, validation etc. Please include a brief flowchart to help readers who are unfamiliar with this topic comprehend the study methodology. I do not see any flowchart for this study, please provide a brief but robust flowchart so that reader can be benefited from the study. Please make a lucrative chart using relevant graphics and symbol. Writing is not scientifically appropriate in this section. Please get some idea from similar articles online. Lines 133-213 are vague please shorten this section and only put relevant information’s. Figure 2 is irrelevant here.

Answer: Section was changed. We use quite standard methodology, same as at the articles cited at the beginning of the section. Figure 2 was changed.

 # Result and discussion

Reviewer: Result sections could be improved at its present form it is hard to understand the finding of the study. As per as the discussion section is concern   This article does not have a proper scientific discussion section from where the reader can get comparative discussion on data and techniques relative to coastal dynamics. If possible please elaborate limitation, future direction and policy implication of this study

Answer: We’ve changed these sections significantly. The main result is the data on retreat rates with the highest possible spatial and temporal resolution. We indicated that we are limited in our analysis by the secrecy of construction data in this area. But the materials presented are also valuable, as they show both the dynamics of the coast in natural conditions and a high vulnerability of this area to changes in the coastal zone.

 # Conclusion

Reviewer: Should be revised, too much of generic sort of discussion here, I cannot see the brief of results of this study here. Please revise conclusion according. Supplementary document is highly desirable otherwise reader would not be benefited. I found 30 % similarity index for this article, I would recommend to authors to rewrite (report attached).

Answer: Conclusions have been corrected. Most of the similarity comes from the conference proceedings that we published recently (POAC) with just a brief information on the topic. We rewrote much part of the quotes.

Reviewer 2 Report

Belova and colleagues provide an integrated analysis based on combined remote sensing and sedimentological proxies for evaluating the coastal erosion of the coastal plain of the Gydan Peninsula. Generally, the topic of the manuscript is of international interest, the work is important, the analysis and methodology are clear enough, and the manuscript is generally well-structured.

More explicitly, all data are sufficient, and the adopted methods are appropriated, as well as the treatment of the data.

The figures are appropriated as both quantity and quality.

The length of the paper is appropriated for this journal, with all interpretations and conclusions to be in general very well justified.

The text is also well organized, and this makes the manuscript easily readable and understandable.

Finally, the bibliography is accurate, without self-citations, and quite updated (with only some additions to be necessary; see minor comments below).

The English is in relatively good shape, but some places need some improvements.

Overall, I have a couple of suggestions/comments, and therefore ask for minor revision before accepting this manuscript for Remote Sensing. So, please take them into account in order this promising contribution to be publishable. The manuscript is acceptable with Minor revision.

Minor Comments and Suggestions:

-L65-66: …”in the conditions of marine sedimentation” …please rephrase it or even better eliminate it

-L70-72: Cite the relevant work of Zarkogiannis et al., 2018

Zarkogiannis S.D., Kontakiotis G., Vousdoukas M.I., Velegrakis A.F., Collins M.B., Antonarakou A., 2018. Scarping of artificially nourished mixed sand and gravel beaches: Sedimentological characteristics of Hayling Island beach, Southern England. Coastal Engineering, 133, 1-12, DOI: 10.1016/j.coastaleng.2017.12.003

Brunier, G., Tamura, T., Anthony, E.J. et al. Evolution of the French Guiana coast from Late Pleistocene to Holocene based on chenier and beach sand dating. Reg Environ Change 22, 122 (2022). https://doi.org/10.1007/s10113-022-01975-3

-L217-222: Split it into 2 different sentences

-L257-259: Briefly explain the reason of the differential retreat rate within the study part of the coast

-L344-345: A reference to the relevant figure is missing here

Author Response

Reviewer: Belova and colleagues provide an integrated analysis based on combined remote sensing and sedimentological proxies for evaluating the coastal erosion of the coastal plain of the Gydan Peninsula. Generally, the topic of the manuscript is of international interest, the work is important, the analysis and methodology are clear enough, and the manuscript is generally well-structured.

More explicitly, all data are sufficient, and the adopted methods are appropriated, as well as the treatment of the data.

The figures are appropriated as both quantity and quality.

The length of the paper is appropriated for this journal, with all interpretations and conclusions to be in general very well justified.

The text is also well organized, and this makes the manuscript easily readable and understandable.

Finally, the bibliography is accurate, without self-citations, and quite updated (with only some additions to be necessary; see minor comments below).

The English is in relatively good shape, but some places need some improvements.

Overall, I have a couple of suggestions/comments, and therefore ask for minor revision before accepting this manuscript for Remote Sensing. So, please take them into account in order this promising contribution to be publishable. The manuscript is acceptable with Minor revision.

Answer: The authors are grateful to the reviewer for a positive assessment of our work. The article was sent to English editing (the one made by the publisher). We tried to take into account all your comments, the remarks on the edits are listed below.

Reviewer: Minor Comments and Suggestions:

-L65-66: …”in the conditions of marine sedimentation” …please rephrase it or even better eliminate it

Answer: We want to emphasize that low-lying coasts with the youngest landforms became a subject of intensive erosion first. We can also exclude the last sentence. Currently changed to:

However, in recent decades, due to global climate change and warming in the Arctic region, the coasts that were previously stable or advancing also began to erode. These are the first coasts formed during the Holocene due to marine sedimentation.

Reviewer: -L70-72: Cite the relevant work of Zarkogiannis et al., 2018

Zarkogiannis S.D., Kontakiotis G., Vousdoukas M.I., Velegrakis A.F., Collins M.B., Antonarakou A., 2018. Scarping of artificially nourished mixed sand and gravel beaches: Sedimentological characteristics of Hayling Island beach, Southern England. Coastal Engineering, 133, 1-12, DOI: 10.1016/j.coastaleng.2017.12.003

Brunier, G., Tamura, T., Anthony, E.J. et al. Evolution of the French Guiana coast from Late Pleistocene to Holocene based on chenier and beach sand dating. Reg Environ Change 22, 122 (2022). https://doi.org/10.1007/s10113-022-01975-3

Answer: Thank you, the references have been added.

Reviewer: -L217-222: Split it into 2 different sentences

Answer: Thank you, the sentence was do too long and difficult to read. Changed to:

«To understand the contribution of different factors to the dynamics of the studied coast and reveal a possible technogenic impact on these dynamics, we first assessed so-called hydrometeorological forcing (HMF) [Ogorodov et al., 2016; Novikova et al., 2018; Shabanova et al., 2018]. This parameter is also called hydrometeorological stress/potential and reflects the combined effect of the changes in the principal hydrometeorological conditions on the coastal dynamics. »

Reviewer: -L257-259: Briefly explain the reason of the differential retreat rate within the study part of the coast

Answer: Changed to (the new text starts directly after the reference to Fig.6):

In the southeastern half of the area, the shoreline retreat rates were less than 1 m/yr. The vegetation line retreated the most (up to -0.9 m/yr) to the northwest of the pier (Figure 6), following the shoreline retreat and due to heavy vehicle passage. We suppose that construction in the central part of the study area is the main reason for the spatial variability of coastal erosion since the coast receded relatively evenly over the entire 7 km before 2014.

Reviewer: -L344-345: A reference to the relevant figure is missing here

Answer: Added the references:

Within this area, traces of sediment removal from the beach and the tide flat are visible in the space images (Fig. 2b, Fig. 3a).

Reviewer 3 Report

This manuscript presents the results of a study examining change in coastline position of an Arctic site on the Gydan Peninsula over different time intervals spanning a record of 48 years using remotely sensed data. I have a number of specific concerns and comments that I share below, but I want to emphasize a few of these with you:

1) Clarity of findings: There needs to be greater effort to separate the hydro-meteorological factors from the human development of the coastline. Of course these are the main drivers of coastline change, but examining a nearby undeveloped portion of the coastline could provide some control for this analysis.

2) Novelty, significance, and rigor of the study: The study focuses on coastline change over time, but needs to be more rooted in the geomorphic processes that are causing this change. Again, it seems clear that the construction of the port has disrupted the coastline. This is not a particularly novel finding, and neither are the methods used to interpret coastline change. How does this study make a significant contribution to the geomorphic study of Arctic coastlines?

3) Grammatical errors: The manuscript is poorly written. There are numerous grammatical errors including, but not limited to tense errors, missing words/articles, unclear phrasing, etc. It would help if the manuscript was written clearly to be better understood. A well-written revision that shows how this work is significant is needed.

Specific comments follow:

Line 39: “The coastal erosion rate is . . .”

Line 55: “Despite the importance of such studies . . .” – Are there others studies? There is reliance on a single source here. If there have been other studies that investigate projected change in the Arctic over the 21st Century, do they conform to the findings of Nielsen et al., 2022?

Lines 60-61: This sentences reads awkwardly to me.

Line 68: “. . . of coastal cliffs, but . . .”

Line 70: “. . . range of research . . .”

Lines 72-73: “. . . study this type of coast in . . .”

Lines 75-79: This thesis statement is brief and needs further development. The background provided in the Introduction leads to this statement, but it falls short of explicitly stating the purpose of this work.

Line 82: “. . . is an elongated gulf in . . .”

Lines 85-87: This sentence is awkwardly constructed and missing articles.

Line 88: “. . . of the coast has been . . .”

Lines 95-96: What is the relevance of this sentence? Does this impact rates of accumulation? What is the geomorphic impact of these spills?

Lines 97-128: Grammatical errors (missing articles and words, incorrect word usage, incorrect tenses, etc.). I am not going to continue to correct grammar. This gets progressively worse through the Materials and Methods section. The entire manuscript needs to be thoroughly proofread.

Lines 128-129: Cite the source that has shown that this is related specifically to global climate change. If other factors contribute to these changes, mention them.

Figure 1: Add territorial names for context. A more detailed map of the actual study site (area within red square) is needed.

Line 144-145: What do you mean by “mark out at the image”?

Figure 2: Scale and an indication of direction/orientation are needed. Indicate what the blue and red lines represent. Also, an inset map showing the location of this site would be useful.

3.2. Analysis of factors influencing coastal dynamics: How was human modification of coastlines and human impact on erosional/depositional processes analyzed? How was the impact of global climate change analyzed?

Figure 3: Indicate which image is a) and b). Indicate what the yellow and red lines represent.

Line 256: This is the first mention of a port. The human modification of the coastline/landscape needs to be discussed before this point of the paper.

Line 273: Was variation in sediment transport by the Khaltsanayakha River considered as a factor impacting the coastline near its mouth?

Lines 281-282: Seeing that shoreline retreat was greatest around the mouths of rivers, their dynamics and consequent coastal impacts need to be explored. Are these rivers gaged? Are there stream discharge records?

Lines 291, 296: It seems that much of the change is framed relative to the location of the mouths of rivers. The rivers need to be discussed in the description of your study area, and their impact on coastal dynamics needs to be investigated.

Figure 4: Indicate which image is a) and b). An indication of direction/orientation is needed. Identify the rivers.

Figures 5 and 6: The shoreline and vegetation lines need to be darker/bolder. The lines are faint and difficult to see.

Figures 7 and 8: Beginning is misspelled above the image to the right. Indicate which image is a) and b). The rate of change on the image to the left is difficult to see/interpret when viewed at 100%.

Lines 364-365: The development of the port and infrastructure has clearly disrupted the natural processes occurring along the coastline. How is this finding significant or novel?

Figure 9 and 10: What are the values of the y axes?

Line 432: Cite these reports.

Lines 456-460: How can you separate hydro-meteorological forcing from the disruption caused by construction and land development? Perhaps you should consider similar analysis of an undeveloped site in the region to establish a baseline or control for how hydro-meteorological factors impact adjacent coastlines.

Lines 500-502: Does this conflict with the warming projections (Pörtner, 2019) you mentioned at the beginning of the paragraph? Will warming not lead to sea-level rise and coastal retreat?

5.4. Compare to other Arctic regions: Comparison is the right idea; however, it needs to be approached differently. There are too many factors that control shoreline advancement/retreat to meaningfully compare rates across such wide geographic areas. That being said, there is value in examining the processes that are leading to these trends. Is there a common thread amongst these processes identified in other studies, or similarity in the main geomorphic or human factors controlling shoreline advancement/retreat? If so, this may lead credence to validation of global climate change at work. But as presented in this study, I do not see how you can definitively make any claim about the impact of climate change.

Line 523: “. . . Bering Land Bridge National Preserve . . . “

Author Response

Reviewer: This manuscript presents the results of a study examining change in coastline position of an Arctic site on the Gydan Peninsula over different time intervals spanning a record of 48 years using remotely sensed data. I have a number of specific concerns and comments that I share below, but I want to emphasize a few of these with you:

1) Clarity of findings: There needs to be greater effort to separate the hydro-meteorological factors from the human development of the coastline. Of course these are the main drivers of coastline change, but examining a nearby undeveloped portion of the coastline could provide some control for this analysis.

Answer: We agree with the reviewer on the need to study adjacent coastal segments for a clearer separation of natural / anthropogenic factors contribution. When we started our work, we did not expect that the coastal retreat zone would be so significant and that the segments of the coast adjacent to the rivers’ mouths would represent natural dynamics. For further analysis, we are now ordering detailed images to the north and south of the surveyed areas. It was difficult to separate hydrometeorological factors from anthropogenic ones, as for information about the construction details is not public (we indicated that in the corrected text of the article). Nevertheless, it was possible to establish the direction of natural processes based on studying of the period before the construction, and it can also be indirectly judged by the dynamics of the coastal segments most distant from the construction sites (primarily in the southeast, where a smaller river flows into the Gulf of Ob, so its influence on the dynamics is less than at the north-west part).

Reviewer: 2) Novelty, significance, and rigor of the study: The study focuses on coastline change over time, but needs to be more rooted in the geomorphic processes that are causing this change. Again, it seems clear that the construction of the port has disrupted the coastline. This is not a particularly novel finding, and neither are the methods used to interpret coastline change. How does this study make a significant contribution to the geomorphic study of Arctic coastlines?

Answer: We tried to indicate the importance of the work in the text. The scale of the coastline changes due to construction may vary significantly. Added to the text of the article: “Relatively stable coasts of the Gulf of Ob with similar rates of pro-gradation/retreat observed in recent decades differ in resistance to technogenic impact and climate changes. For example, the construction of the Sabetta port on the opposite side of the Gulf of Ob, the eastern coast of Yamal Peninsula 65 km away from the study site, has not led to such a sharp increase in retreat rates. We suppose that this is due to the difference in natural conditions, namely, nearshore bathymetry and longshore sediment transport.”

Reviewer: 3) Grammatical errors: The manuscript is poorly written. There are numerous grammatical errors including, but not limited to tense errors, missing words/articles, unclear phrasing, etc. It would help if the manuscript was written clearly to be better understood. A well-written revision that shows how this work is significant is needed.

Answer: We really worked hard on this topic and missed many obvious mistakes by editing the same text many times. The corrected text of the article was sent to English editing made by the publisher.

Reviewer: Specific comments follow:

Line 39: “The coastal erosion rate is . . .”

Answer: The sentence has been corrected

Reviewer: Line 55: “Despite the importance of such studies . . .” – Are there others studies? There is reliance on a single source here. If there have been other studies that investigate projected change in the Arctic over the 21st Century, do they conform to the findings of Nielsen et al., 2022?

Answer: Corrected. There are no other similar studies, since forecasting the dynamics of the Arctic coasts must take into account a lot of factors (both regional and global), and the coasts themselves are heterogeneous in structure.

Reviewer: Lines 60-61: This sentences reads awkwardly to me.

Answer: We agree, these sentences do reads badly, changed to:

Most of the studies of Arctic coasts are focused on the dynamics of thermo-abrasional coasts [Baranskaya et al., 2021, Belova et al., 2020, Jones et al., 2018, Novikova et al., 2018, Gibbs et al., 2021, and others]. This type of coast has a pronounced coastal cliff, in which erosion and destruction lead to coastal retreat.

Reviewer: Line 68: “. . . of coastal cliffs, but . . .”

Answer: Corrected.

Reviewer: Line 70: “. . . range of research . . .”

Answer: Corrected.

Reviewer: Lines 72-73: “. . . study this type of coast in . . .”

Answer: Corrected.

Reviewer: Lines 75-79: This thesis statement is brief and needs further development. The background provided in the Introduction leads to this statement, but it falls short of explicitly stating the purpose of this work.

Answer: Changed the introduction section above. The following paragraph added above:

“Human activity on the Arctic coast is constantly expanding. From 2000 to 2020, the area of built-up territories within the Arctic coastal zone increased by 15% [Bartsch et al., 2021], with the main increase observed in Russia. For Arctic settlements, the risks of coastal retreat caused by climate change will increase in the 21st century [Constable et al., 2022]. Within the settlements, it is often difficult to separate the natural factors of coastal dynamics activation from the anthropogenic ones.”

Reviewer: Line 82: “. . . is an elongated gulf in . . .”

Answer: Corrected.

Reviewer: Lines 85-87: This sentence is awkwardly constructed and missing articles.

Answer: Excluded this sentence (the information was moved to other parts of the text).

Reviewer: Line 88: “. . . of the coast has been . . .”

Answer: Corrected.

Reviewer: Lines 95-96: What is the relevance of this sentence? Does this impact rates of accumulation? What is the geomorphic impact of these spills?

Answer: The sentence was moved to the Introduction with some additions (as proof of the importance of studying of the low-lying Arctic coasts):

“Accumulative low-lying coasts are highly sensitive to oil and oil product spills [Petersen et al., 2002, Kara Sea..., 2016]; thus, the study of their dynamics is of high importance for the industry.”

Reviewer: Lines 97-128: Grammatical errors (missing articles and words, incorrect word usage, incorrect tenses, etc.). I am not going to continue to correct grammar. This gets progressively worse through the Materials and Methods section. The entire manuscript needs to be thoroughly proofread.

Answer: The corrected text of the article was sent to English editing made by the publisher.

Reviewer: Lines 128-129: Cite the source that has shown that this is related specifically to global climate change. If other factors contribute to these changes, mention them.

Answer: Corrected:

“In recent decades, there has been a trend of rising summer temperatures, prolonged ice-free period duration in the sea [Shabanov, 2022], and an increase in storm frequency due to global climate change [Ogorodov et al., 2016].”

Reviewer: Figure 1: Add territorial names for context. A more detailed map of the actual study site (area within red square) is needed.

Answer: The study area map was partly corrected. The whole study area can be seen at figures with results, we decided not to duplicate it.

Reviewer: Line 144-145: What do you mean by “mark out at the image”?

Answer: Corrected:

“Shoreline property—sea–land border. As a rule, this is easy to identify in an image;”

Reviewer: Figure 2: Scale and an indication of direction/orientation are needed. Indicate what the blue and red lines represent. Also, an inset map showing the location of this site would be useful.

Answer: The figure was remade.

Reviewer: 3.2. Analysis of factors influencing coastal dynamics: How was human modification of coastlines and human impact on erosional/depositional processes analyzed? How was the impact of global climate change analyzed?

Answer: We’ve added the following paragraph to the end of section 3.2:

“Human activity has influenced the coastal dynamics of the studied section since the beginning of port facilities’ construction in 2014. The details of the project’s construction (including the direct impact on the coastal zone) are restricted corporate data. Accordingly, in this article, we can only partly analyze the role of technogenic factors based on open data, such as satellite imagery. For example, the expansion of the built-up area in the central part of the studied coast can be seen in the images. However, the impact of construction is not limited to the areas of artificial coasts since the dynamics of coastal segments without constructions are also affected by the removal of sands from the beach and tide flat, or by the dredging inevitable during port construction.”

Reviewer: Figure 3: Indicate which image is a) and b). Indicate what the yellow and red lines represent.

Answer: Corrected.

Reviewer: Line 256: This is the first mention of a port. The human modification of the coastline/landscape needs to be discussed before this point of the paper.

Answer: Corrected (the paragraph added to section 3.2).

Reviewer: Line 273: Was variation in sediment transport by the Khaltsanayakha River considered as a factor impacting the coastline near its mouth?

Answer: Thank you, this is a valuable comment. We have no data on sediment transport of Khaltsanayakha River. Most likely, there were no such works. What we’ve done by now – we’ve added a paragraph describing the two rivers on the boundaries of the study area.

“The rivers on the borders of the studied coast serve as an additional sediment source for the coastal zone. The 0.9 km long beach ridge south of the mouth of the Khaltsanayakha River is currently subject to erosion. From 1972 to the present, no spits have been formed at the river mouths. At the given intensity of longshore sediment transport, the material brought by the rivers is insufficient for the formation of accumulative forms. The Khaltsanayakha River limits the study area from the northwest. The 55 km long meandering river has a catchment area of 210 km2; its banks are mainly composed of sandy soils. The Nadyaipingche River, which bounds the studied coast from the southeast, is much smaller—its length is 20 km, and the catchment area is 65 km2.”

Reviewer: Lines 281-282: Seeing that shoreline retreat was greatest around the mouths of rivers, their dynamics and consequent coastal impacts need to be explored. Are these rivers gaged? Are there stream discharge records?

Answer: We agree that these two rivers, especially the Khaltsanayakha River on the northwest of the study area, need to be considered as a factor influencing coastal dynamics. We’ve found no data on river discharges (though we have an access to some reports). We included a paragraph into the section 5.3:

“The construction has changed the sediment transport of the Khaltsanayakha River. The 2020 image shows a temporary bridge 4 km upstream from the river mouth with wide embankments on both banks of the river. Muddy river water indicates partial erosion of this bridge. Without a special study, it is impossible to say whether the supply of material to the coastal zone has increased or decreased. On the one hand, the bridge could reduce the amount of sediments transported to the river mouth; as well as the extraction of sand from the riverbed. On the other hand, erosion of bridge embankments causes additional sediment supply.”

Reviewer: Lines 291, 296: It seems that much of the change is framed relative to the location of the mouths of rivers. The rivers need to be discussed in the description of your study area, and their impact on coastal dynamics needs to be investigated.

Answer: Thank you, we’ve added the description to the Study area section (see the comment above).

Reviewer: Figure 4: Indicate which image is a) and b). An indication of direction/orientation is needed. Identify the rivers.

Answer: Corrected.

Reviewer: Figures 5 and 6: The shoreline and vegetation lines need to be darker/bolder. The lines are faint and difficult to see.

Answer: Corrected.

Reviewer: Figures 7 and 8: Beginning is misspelled above the image to the right. Indicate which image is a) and b). The rate of change on the image to the left is difficult to see/interpret when viewed at 100%.

Answer: Corrected.

Reviewer: Lines 364-365: The development of the port and infrastructure has clearly disrupted the natural processes occurring along the coastline. How is this finding significant or novel?

Answer: A paragraph was added to Results: “Relatively stable coasts of the Gulf of Ob with similar rates of pro-gradation/retreat observed in recent decades differ in resistance to technogenic impact and climate changes. For example, the construction of the Sabetta port on the opposite side of the Gulf of Ob, the eastern coast of Yamal Peninsula 65 km away from the study site, has not led to such a sharp increase in retreat rates. We suppose that this is due to the difference in natural conditions, namely, nearshore bathymetry and longshore sediment transport.”

A paragraph was added to Conclusions:

The investigated eastern coast of the Gulf of Ob is more vulnerable to changes in both the natural environment (sea level rise, wave regime, etc.) and technogenic impact (construction in the coastal zone) compared to neighboring areas of technogenic development. Under changing climate, it is likely that coastal retreat here will continue.

Reviewer: Figure 9 and 10: What are the values of the y axes?

Answer: The values have been added to the figures’ captions

Reviewer: Line 432: Cite these reports.

Answer: Thank you, we’ve missed that. We’ve cited a publication in Russian about coastal dynamics of the Gulf of Ob (both here and in the Study area section).

Reviewer: Lines 456-460: How can you separate hydro-meteorological forcing from the disruption caused by construction and land development? Perhaps you should consider similar analysis of an undeveloped site in the region to establish a baseline or control for how hydro-meteorological factors impact adjacent coastlines.

Answer: Yes, that is what we plan to do (answered this comment earlier, see the 1st comment).

Reviewer: Lines 500-502: Does this conflict with the warming projections (Pörtner, 2019) you mentioned at the beginning of the paragraph? Will warming not lead to sea-level rise and coastal retreat?

Answer: Thank you, we should emphasize that coastal process erosion will continue due to climate change, only short-time stabilization is possible (and that wouldn’t mean the erosion process has stopped). We’ve added the following to the end of section 5.3:

“But if we consider longer time period, the erosion will go on due to sea level rise and other consequences of climate change such as longer ice-free conditions and more frequent storms.”

Reviewer: 5.4. Compare to other Arctic regions: Comparison is the right idea; however, it needs to be approached differently. There are too many factors that control shoreline advancement/retreat to meaningfully compare rates across such wide geographic areas. That being said, there is value in examining the processes that are leading to these trends. Is there a common thread amongst these processes identified in other studies, or similarity in the main geomorphic or human factors controlling shoreline advancement/retreat? If so, this may lead credence to validation of global climate change at work. But as presented in this study, I do not see how you can definitively make any claim about the impact of climate change.

Answer: We agree with your comment. We left this paragraph to illustrate the observed trends. In fact, the diversity of coasts, hydrometeorological conditions, human impact always make the comparison of coastal dynamics quite difficult.

Reviewer: Line 523: “. . . Bering Land Bridge National Preserve . . . “

Answer: Corrected.

Reviewer 4 Report

Dear Authors,

It was a pleasure to review your submitted manuscript and gathering information on shoreline retreat in polar marine environments is never an easy task. I congratulate you on doing this well! I am also certain that these data will be used more widely in other regions around the Arctic Circle. I’ve recommended minor revisions and most of these are indicated on an annotated below which accompanies this review.

Broadly, I recommend that you put an paragraph on Assessing the Vulnerability of Coasts to Climate Change, in accordance with articles published by Abuodha and Woodroffe (2007), Thieler and Hammar-Klose (1999) and Special Report on the Ocean and Cryosphere in Change (2019). See references below:

·         ABUODHA, P. A.; WOODROFFE, C. D. Assessing Vulnerability of Coasts to Climate Change: A Review of Approaches and Their Application to the Australian Coast. In: WOODROFFE, C. D.; BRUCE, E.; PUOTINEN, M.; FURNESS, R. A. (Ed.). GIS for the Coastal Zone: A Selection of Papers from CoastGIS 2006. 1ª Ed. Wollongong: Australia National Centre for Ocean Resources and Security, 2007. 458p.

·         THIELER, E. R.; HAMMAR-KLOSE, E. S. National Assessment of Coastal Vulnerability to Sea-Level Rise: Preliminary Results for the U.S. Pacific Coast. U.S. Geological Survey Open-File Report, v. 00-178, p. 1-17, 1999.

·         SPECIAL REPORT ON THE OCEAN AND CRYOSPHERE IN A CHANGING. Chapter 4: Sea Level Rise and Implications for Low Lying Islands, Coasts and Communities. IPCC - SR Ocean and Cryosphere (SROCC), p. 1-169, 2019.

The figure 2 (a, b, c and d) are low resolution. We recommend improving the quality and metric scale.

We recommend that all figures and tables are positioned after their citation paragraph.

The conclusions should reflect the objectives of manuscript. It was very concise. We believe it can improve. We recommend a short of English language review.

We recommend minor modifications so that the manuscript can be published.

Sincerely,

Author Response

Reviewer: Dear Authors,
It was a pleasure to review your submitted manuscript and gathering information on shoreline retreat in polar marine environments is never an easy task. I congratulate you on doing this well! I am also certain that these data will be used more widely in other regions around the Arctic Circle. I’ve recommended minor revisions and most of these are indicated on an annotated below which accompanies this review.

Answer: Dear Reviewer,

We appreciate your kind response to the work we’ve done!

Reviewer: Broadly, I recommend that you put an paragraph on Assessing the Vulnerability of Coasts to Climate Change, in accordance with articles published by Abuodha and Woodroffe (2007), Thieler and Hammar-Klose (1999) and Special Report on the Ocean and Cryosphere in Change (2019). See references below:

  • ABUODHA, P. A.; WOODROFFE, C. D. Assessing Vulnerability of Coasts to Climate Change: A Review of Approaches and Their Application to the Australian Coast. In: WOODROFFE, C. D.; BRUCE, E.; PUOTINEN, M.; FURNESS, R. A. (Ed.). GIS for the Coastal Zone: A Selection of Papers from CoastGIS 2006. 1ª Ed. Wollongong: Australia National Centre for Ocean Resources and Security, 2007. 458p.
  • THIELER, E. R.; HAMMAR-KLOSE, E. S. National Assessment of Coastal Vulnerability to Sea-Level Rise: Preliminary Results for the U.S. Pacific Coast. U.S. Geological Survey Open-File Report, v. 00-178, p. 1-17, 1999.
  • SPECIAL REPORT ON THE OCEAN AND CRYOSPHERE IN A CHANGING. Chapter 4: Sea Level Rise and Implications for Low Lying Islands, Coasts and Communities. IPCC – SR Ocean and Cryosphere (SROCC), p. 1-169, 2019.

Answer: Thank you for this advice. We’ve added a few sentences at the end of paragraph 5.3, citing the abovementioned publications:

“Climate change will lead to coastal retreat [Special report…, 2019]. It is of great importance to assess the vulnerability of the studied coasts to the changing climate, as it was carried out for other regions of the world [Abuodha et al., 2007; Thieler et al., 1999]. This will help develop an adaptation strategy of human activity to the hazards associated with sea level changes.”

In other parts of the article, we also indicated that the erosion of the studied coast will continue with the sea level rise.

Reviewer: The figure 2 (a, b, c and d) are low resolution. We recommend improving the quality and metric scale.

Answer: The figure was remade.

Reviewer: We recommend that all figures and tables are positioned after their citation paragraph.

Answer: Corrected

Reviewer: The conclusions should reflect the objectives of manuscript. It was very concise. We believe it can improve. We recommend a short of English language review.

Answer: Thank you, we changed the conclusions – added the 2nd paragraph and edited another two:

“The studied section of the Gulf of Ob coast was stable or accumulative until 2014 along all 7 km of the coast. In recent decades, enhanced hydrometeorological forcing and the construction of port facilities have led to considerable changes in the lithodynamic processes. In 2014-2017 the coastline retreated at an average rate of -5.8 m/ yr, and in 2017-2020, its average rate was -3.8 m/yr. The maximal rates of re-treat took place near the new constructions. In the coming years, with Arctic climate warming and continued human impact, erosion of the studied coast may intensify.

The investigated eastern coast of the Gulf of Ob is more vulnerable to changes in both the natural environment (sea level rise, wave regime, etc.) and technogenic impact (construction in the coastal zone) compared to neighboring areas of technogenic development. Under changing climate, it is highly probable that coastal retreat will continue.

For engineering work at all stages from design to construction and operation, we recommend considering probable coastal retreat; identifying infrastructure that may be adversely impacted; determining primary factors driving active erosion (earthworks, deposits drift, etc.); and monitoring the coastal retreat to prepare potential mitigation measures for coastal infrastructure.”

The corrected text of the article was sent to English editing made by the publisher.

Round 2

Reviewer 1 Report

Authors made a significant amount of revision, i'm happy with the revision